# Symmetry breaking in optimal transport networks

Siddharth Patwardhan [1], Marc Barthelemy[2,3] ✉, Şirag Erkol[4], Santo Fortunato[1] ✉ & Filippo Radicchi [1] ✉

Engineering multilayer networks that efficiently connect sets of points in space is a crucial task in all practical applications that concern the transport of people or the delivery of goods. Unfortunately, our current theoretical understanding of the shape of such optimal transport networks is quite limited. Not much is known about how the topology of the optimal network changes as a function of its size, the relative efficiency of its layers, and the cost of switching between layers. Here, we show that optimal networks undergo sharp transitions from symmetric to asymmetric shapes, indicating that it is sometimes better to avoid serving a whole area to save on switching costs. Also, we analyze the real transportation networks of the cities of Atlanta, Boston, and Toronto using our theoretical framework and find that they are farther away from their optimal shapes as traffic congestion increases.

Networks that provide optimal transport properties[1,2] are of interest in many different disciplines ranging from the study of natural systems such as water transport in plants[3], veination patterns in leaves[4,5], river basins[6] to the design of transportation infrastructures, either from an applied point of view[7], or from a more mathematical perspective[8]. In particular, transportation networks evolve in time and their structure has been studied in many contexts from street networks to railways and subways[9–15]. The evolution of transportation networks is also relevant to biological cases such as the growth of slime mould[16] or social insects[17,18].

An important problem consists in designing a network from scratch or extending an existing network; this is a central subject in transportation and location science, usually known as the network design problem[19]. Such a problem, applied to rapid transit networks, for example, is divided into three sub-problems, which are solved numerically: location of new stations, construction of the core network connecting these stations, and location of intermediate stations on the network. From an engineering point of view, this type of problem can be solved with various optimization methods on practical cases, but the general behavior of optimal solutions is not known. From a purely mathematical point of view, there have been extensive studies of optimal networks over a given set of nodes (such as the minimum spanning tree[20], or other optimal trees[21]). Some of these problems allow for extra chosen nodes such as the Steiner tree problem[22], or geometric location problems in which demand points are to be matched with supply points[23]. Another example is the much-studied Monge-Kantorovich mass transportation problem[24], involving matching points from one distribution with points from another distribution.

The main problem in network design is fundamentally different. We are given the density of population and we are looking for the network that minimizes some objective function involving some average time, in general (although other choices are possible, see for example[7]). In this setting, there are usually two different transport modes, a slow one representing for example cars on the road network, and a fast one representing the subway or some rapid transit network. The natural framework here is then the one of multiplex networks comprising two different transportation networks, one known while the structure of the second one is to be determined (for multiplexes in the context of optimization see for example[25]). A practical realization of this problem concerns the specific case of subways (for a network analysis of subways, see for example[10,26–34]). In most large cities, a subway system has been built and later enlarged, with current total

[1]Center for Complex Networks and Systems Research, Luddy School of Informatics, Computing, and Engineering, Indiana University, Bloomington, IN 47408, USA. [2]Université Paris-Saclay, CNRS, CEA, Institut de Physique Théorique, 91191 Gif-sur-Yvette, France. [3]Centre d'Analyse et de Mathématique Sociales (CNRS/EHESS) 54 Avenue de Raspail, 75006 Paris, France. [4]Center for Science of Science and Innovation, Kellogg School of Management, Northwestern University, Evanston, IL 60208, USA. ✉e-mail: marc.barthelemy@gmail.com; santo@indiana.edu; filiradi@indiana.edu

lengths varying from a few kilometers to a few hundred kilometers. The geometry of these networks, as its total length increases, varies from simple lines to more complex shapes with loops for larger networks[35,36]. In particular, for the largest networks, convergence to a structure with a well-connected central core and branches reaching out to suburbs has been observed[10].

Algorithmic aspects of network design have been studied within computational geometry (e.g.,[37] chapter 9) and location science (e.g.,[7] and references therein), and some simpler problems of this type have been addressed previously. For instance, the problem of the quickest access between an area and a given point was discussed in[38,39]. In network science, the optimization problem is traditionally recast as a navigation problem in lattices with long-range connections[40,41]. However, our specific question – optimal network topologies as a function of population distribution and network length – is largely an open problem. In[36], some results were obtained in two-dimensional systems by comparing a priori defined optimal network configurations. First, it was shown that, if the goal is reaching a single point in the plane, then the optimal network is necessarily a tree. Second, the paper hinted at the possibility of the existence of transitions between optimal configurations when the length of the network changes. More precisely, it has been shown that as the length of the network increases resources go preferentially to radial branches and that there is a sharp transition at a critical value of the length where a loop appears.

In this paper, we address the problem of the quickest average time to access a central point using a multiplex framework (see Fig. 1). We are given the structure of one layer, and we are allowed to build an additional layer that can facilitate quicker access to the central point. The new layer is characterized by a faster speed than the existing, slow layer; however, changes of layers incur a cost. We study the optimization problem of finding the best configuration of the fast layer on systems of arbitrary dimensions. We solve exactly the optimization problem for one-dimensional systems, showing that the optimal fast-layer configuration undergoes a sharp transition between a perfectly symmetric configuration and a fully asymmetric configuration. We numerically show that such symmetry breaking in optimal networks occurs in systems of arbitrary dimensions. We specifically focus on two-dimensional lattices and perform a systematic study of transportation systems within real cities, where we use the slow layer to model the road network and the fast layer to model the subway network. We find that real subways display network topologies compatible with the optimal ones that can be obtained using our computational framework. Differences between real and optimal networks typically arise as the ratio between subway and car speeds increases.

## Results

### Multiplex transportation model

We consider a well-established network model for multimodal transportation systems (see Methods and Supplementary Information Section 1 for details)[12,42,43]. The model comprises two network layers, namely a slow layer $\mathcal{G}$ whose set of edges is denoted with $\mathcal{S}$ and a fast layer $\mathcal{H}$ whose set of edges is $\mathcal{F}$. The layers denote different modes of transportation, e.g., cars and subways. Each node $n$ in the slow layer has a mirror image, or replica, in the fast layer $F(n)$. For example, we can think of node $n$ in the slow layer as an intersection between roads, and of $F(n)$ in the fast layer as the subway station corresponding to that intersection. The system is such that edges in the fast layer are a subset of the replica edges of the slow layer; essentially, not all roads are mirrored by subway segments. We assume that edges in the two layers are traversed at different speeds, and without loss of generality, we parameterize the speed ratio with $0 \leq \eta \leq 1$. Agents departing from nodes in the slow layer move along their quickest path towards a specific node $o$ in the slow layer, the so-called center of the network. These quickest paths can involve edges in both layers; however, each change of layer, happening between replica nodes, has a cost equal to $c \geq 0$. See

Fig. 1 for a schematic example. For a given configuration $\mathcal{F}$ of the fast layer, we can find the minimum-cost path of each node $n$ in the slow layer to the center $o$. We then measure the efficiency of $\mathcal{F}$ in terms of the average time to reach the center, i.e.,

$$\tau(\mathcal{F}) = \frac{\sum_{n \in \mathcal{G}} d_n p_n}{\sum_{n \in \mathcal{G}} p_n} . \tag{1}$$

Here, $d_n$ is the cost of the fastest path of the generic node $n$ to $o$. Also, we assume that each node $n$ in the slow layer has an associated weight or demand $p_n \geq 0$. In a real city, $p_n$ is proportional to the actual density of population associated with node $n$. $\tau(\mathcal{F})$ is computed over all nodes in the slow layer only, but eventual minimum-cost paths can take advantage of edges in the fast layer.

The goal of our modeling framework is finding the best or optimal configuration $\mathcal{F}^*$ of the fast layer, i.e., the configuration that corresponds to the minimum value of the average time to reach the center starting from the nodes of the slow layer. The optimization problem defined in Eq. (4) is constrained by the number of edges $L$ can be used to form the fast layer. Note that $L$ is interpreted as the cost of building the fast layer, hence is measured in the same units as $\tau$ and $c$. We are interested in providing a full characterization of the topology of the optimal fast layer as a function of the parameters $\eta$ and $c$ of the multiplex transportation model. We study this optimization problem under different settings determined by the topology of the slow layer.

### Symmetry breaking

We begin our investigation by studying a one-dimensional version of our model (see Methods for details). For simplicity, the model is thought in continuous space. However, the calculations and results can be immediately generalized to a one-dimensional discrete lattice. The slow layer consists of a segment of length $2R$ extending symmetrically around the origin or center $o$. In the computation of the continuous version of the objective function of Eq. (1), we further assume that all parts of the slow layer have equal weight, i.e., $p_n = \text{const}$. An illustration of the system is shown in Fig. 2.

We remind that the problem is finding the optimal configuration of the fast layer such that the average cost to reach the center from any point of the slow layer is minimum (see Eqs. (1) and (4)). The optimization problem is constrained by the fact that the fast layer has a fixed cost $L$, with $L \leq R$. We mathematically prove that the topology of the optimal fast layer undergoes a series of phase transitions depending on the values of the model parameters $L$, $\eta$, and $c$.

A first, trivial critical point is given by $r_c = 2c/(1 - \eta)$ (see also Eq. (3) in "Methods"): there is no advantage in having a fast layer with length $L \leq r_c$, as a fast layer with such a cost is not used in any minimum-cost paths to the center. The optimization problem is then subject to the constraint that the fast layer should be of length $r_c \leq L \leq R$.

As we prove in the Supplementary Information Section 3, solutions to this optimization problem are given by connected segments that include the replica of the center $F(o)$. We can then parameterize the optimal fast layer by a single quantity $0 \leq \alpha \leq 1/2$, such that the fast layer extends over a length $\alpha L$ to the right of $F(o)$ and over $(1 - \alpha)L$ to the left of $F(o)$. We find that only two configurations for the optimal fast layer are possible: (i) a completely asymmetric configuration obtained for $\alpha = \alpha^* = 0$ (when $L \leq L^\dagger$); (ii) a completely symmetric configuration obtained for $\alpha = \alpha^* = 1/2$ (when $L \geq L^\dagger$). The critical value $L^\dagger$ where the transition occurs is

$$L^\dagger = \sqrt{4Rr_c - 2r_c^2} . \tag{2}$$

Typical phase diagrams are displayed in Fig. 2. We clearly see a discontinuous transition between the symmetric and the asymmetric optimal configurations as the parameters of the model vary. This is a rather surprising result as it indicates that, under certain

circumstances, the optimal solution is obtained by constructing a fast layer only on one side of the system. In other circumstances instead, a symmetric configuration is more advantageous than the asymmetric one.

The physical intuition behind this curious behavior is as follows. Constructing a fast layer requires an initial waste of resources, as only parts of the slow layer whose minimum cost to reach the center is at least $r_c$ take effective advantage of the fast layer. Such an initial investment consists of building a fast layer such that $L \geq r_c$ for the asymmetric case, but $L \geq 2r_c$ for the symmetric configuration. Hence, for $r_c \leq L \leq 2r_c$, the asymmetric configuration is trivially preferred over the symmetric one; however, the situation is not immediately inverted for $L \geq 2r_c$. As a matter of fact, any further extension of a branch of the fast layer leads to a reduction of the time to reach the center for all parts of the slow layer that are served by that branch. However, the objective function is subject to a diminishing return as the branch of the fast

layer grows towards the boundary of the system. As $L$ increases, the only branch of the asymmetric configuration grows twice as fast towards the boundary than the two branches of the symmetric configuration. Thus, the initial advantage of building an asymmetric fast layer over a symmetric one is still present for $L \geq 2r_c$, but the gap narrows as the size $L$ of the fast layer increases. The critical value $L^\dagger$ of Eq. (2) denotes the size of the fast layer when the two configurations generate identical reduction in travel time to the center and, for $L \geq L^\dagger$ the symmetric configuration is preferred over the asymmetric one. The diminishing-return property of the objective function explains also why the optimal configuration for $L \geq L^\dagger$ must be symmetric. If we alter in fact the symmetric configuration by reducing one branch in favor of the other, then the increase of the objective function induced by the reduction of the one branch will be larger than the decrease of the objective function induced by the extension of the other branch. Hence, by altering the symmetric configuration, we will necessarily increase the average time to reach the center for the overall system.

As we prove in the Supplementary Information Section 6, symmetry breaking occurs not only for $p_n = \text{const.}$, but for arbitrary functions $p_n$ that are symmetric with respect to the center of the network; however, the value of the transition point between the symmetric and the asymmetric phases depends on the specific functional form of $p_n$.

Although we have mathematical support for the above interpretation only in one-dimensional systems, we believe that the general principle of symmetry breaking applies to any network regardless of the dimension of the space where the network is embedded. Indeed, in our numerical experiments we do observe symmetry breaking in the geometry of the optimal configuration of the fast layer also in systems with dimension $d > 1$. We discuss these findings below.

We first extend our analysis to two-dimensional triangular lattices. The center $o$ of the slow layer is identified by the site corresponding to the geometric center of the lattice and all other nodes in the layer are identified by lattice sites at distance at most $R$ from such a center, see Methods for details. Due to the computational complexity of the optimization problem of Eq. (4), optimal configurations of the fast layer cannot be determined exactly in this case. We rely instead on the greedy optimization strategy described in the Supplementary Information Section 4. The submodularity of the objective function of Eq. (1) that we prove in the Supplementary Information Section 4 allows us to use this algorithm to generate, in a time that roughly grows as $R^{2.7}$, approximate solutions to the optimization problem of Eq. (4) that are at most a factor $(1 - 1/e) \simeq 0.63$ above the ground-truth minimum[44].

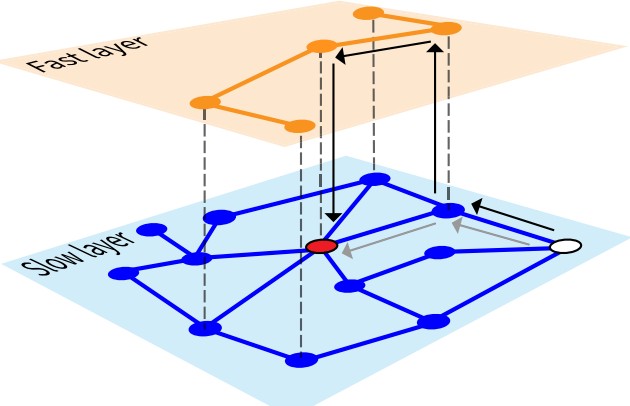

**Fig. 1 | Illustration of the multiplex transportation model.** In the slow layer (blue), the time required to traverse an edge equals one; in the fast layer (orange), each edge is traversed in a time reduced by a factor $0 \leq \eta \leq 1$. Replica nodes across layers are connected by edges whose transit time is $c \geq 0$ denoted by the dashed segments. Two possible paths connecting the white node to the center node given in red are shown. The path shown by gray arrows requires a total time equal to 2 as it uses only two edges in the slow layer. The second path, highlighted by black arrows, involves two changes of layer, one edge in the fast layer, and one edge in the slow layer, resulting in a total transit time equal to $1 + \eta + 2c$. The second path is faster than the first one as long as $1 + \eta + 2c < 2$.

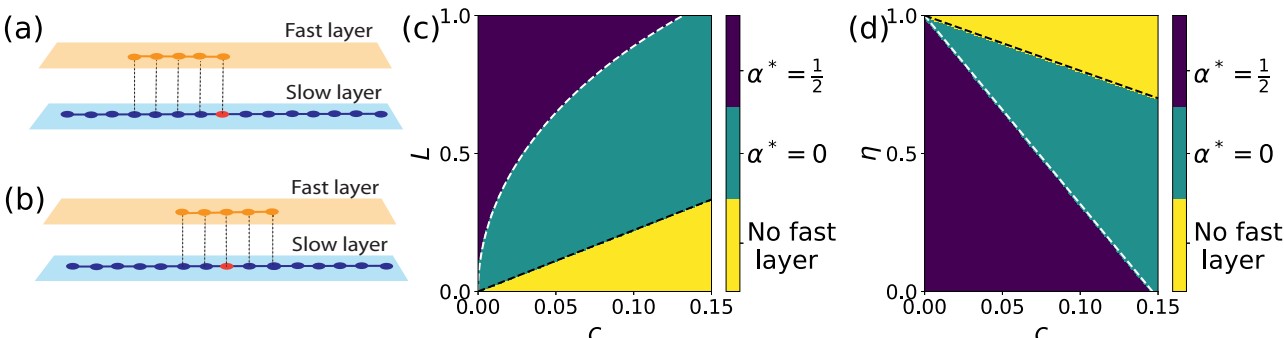

**Fig. 2 | Symmetry breaking in one-dimensional systems. a, b** We consider a slow layer given by a segment of length $2R$ that extends symmetrically around its center (red circle). The fast layer is given by a segment of length $L$, with a portion of length $\alpha L$ on the right of the center and a portion of length $(1 - \alpha)L$ on the left of the center, where $0 \leq \alpha \leq 1/2$. In the figure, we use $R = 7$ and $L = 4$. As we prove in the Methods section and in the Supplementary Information Section 6, two optimal configurations are possible for the fast layer: **(a)** an asymmetric one, i.e., $\alpha = \alpha^* = 0$, and (b) a symmetric one, i.e., $\alpha = \alpha^* = 1/2$. **c** Optimal configuration of the fast layer as a

function of the model parameter $L$ and the switching cost $c$. Here the ratio of the speeds of the fast and the slow layers is $\eta = 0.1$. We distinguish three regions: (i) for $L \leq r_c$, with $r_c = 2c/(1 - \eta)$ as defined in Eq. (3) and represented by the dashed black line, the fast layer is not used; (ii) for $r_c \leq L \leq L^\dagger$, with $L^\dagger$ defined in Eq. (2) and represented by the white dashed curve, then $\alpha^* = 0$; (iii) for $L \geq L^\dagger$, then $\alpha^* = 1/2$. **d** Same as in (**c**), but the optimal geometry of the fast layer is estimated as a function of $c$ and $\eta$ for $L = 1$. The black dashed line is identified by the condition $r_c = L$; the white dashed line is given by Eq. (10).

Typical solutions obtained using greedy optimization are displayed in Fig. 3a, b, and c. Here, we assume that the weight associated with each node $n$ of the slow layer is $p_n$ = const. As discussed in the Methods section and proved in the Supplementary Information Section 3, optimal configurations of the fast layer are given by trees with at least one edge incident to $F(o)$. However, depending on the choice of the model parameters $L$, $\eta$ and $c$, different optimal configurations may emerge. As for the one-dimensional case, also here optimal configurations of the fast layer appear to be characterized by branches of similar length, so that different optimal configurations can be distinguished by simply counting the number of such branches, namely $k^*$ as defined in Eq. (5). We do observe $k^* = 1, 2$ and 3 in Fig. 3a, b and c, respectively. Please note that this simple characterization of the geometry of the fast layer is valid only in the regime $L < R$. For larger sizes of the fast layer, the geometry of the optimal configurations becomes much richer and requires additional order parameters to be described; in this paper, we only consider phase transitions concerning fast layers whose size is much smaller than the one of the slow layer.

Typical phase diagrams are shown in Fig. 3d, e, and f. As for the one-dimensional continuous model, we observe that a fully asymmetric configuration emerges for large $c$ values, see Fig. 3d, e, and f; as $L$ increases, we observe transitions towards larger number of branches, see Fig. 3f. In Fig. 4d, we validate the goodness of the solutions obtained via greedy optimization by comparing them with solutions obtained via simulated-annealing optimization (see Supplementary Information Section 4 for details).

In the Supplementary Information Section 6, we consider a continuous-space approximation of the two-dimensional lattice. The results of our analysis are qualitatively similar to those valid for the discrete lattice, with the only caveat that optimal configurations of the fast layer can be characterized by an unbounded number of branches. In the Supplementary Information Section 5, we also consider two-dimensional lattices where the weight $p_n$ is an exponentially decreasing function of the lattice distance of node $n$ to the center $o$. Results are qualitatively similar to those reported in Fig. 3 in that setting too.

Our findings on the breaking of the symmetry of the optimal fast layer generalize also to infinite-dimensional systems.

In the Supplementary Information Section 6, we consider a continuous-space approximation of a star-like system where the slow layer is given by an arbitrary number $q$ of segments intersecting in a single point and extending symmetrically around this central point. We can prove analytically that the only allowed solutions to our optimization problem are given by fast layers consisting of $1 \le n^* \le q$ branches of identical length $L/n^*$. As in the case of the one- and two-dimensional systems, also for the star-like system we observe that $n^* = 1$ for sufficiently large $c$ values, and that $n^*$ grows as $L$ increases.

The same qualitative behavior is also observed in numerical simulations on one instance of the Erdős-Rényi model with $N = 1000$ nodes and average degree $\langle k \rangle = 4$. For simplicity, in our simulations, we select the node with the largest degree $k_{max} = 11$ as the center of the slow layer. We then determine the optimal configuration of the fast layer via greedy optimization. We observe transitions between configurations of the optimal fast layer with variable number of branches $0 \le k^* \le k_{max}$ depending on the choice of the model parameters. Results of these simulations are reported in the Supplementary Information Fig. 3.

## Real-world cities

In this section, we study the properties of the subway systems in Atlanta, Boston, and Toronto under the lens of our framework. We choose monocentric cities, fairly isolated from other major urban centers, with a tree-like subway structure. We identify the intersection point of the real subway lines in all three cases as the city center. We see that this point corresponds to the downtown area in the three cities.

First, we incorporate real population data in our model. We rely on a two-dimensional triangular lattice multiplex model; we use the population data and the appropriate coordinates reference systems to impose the triangular lattice structure onto the city landscape. Details on the data and modeling of the city population distribution can be found in the Methods section. We denote all quantities relative to the real physical system using the same notation as for the multiplex model, but we add a tilde on top of the corresponding symbol. For example, $R$ indicates the radius of the lattice model, and $\tilde{R}$ denotes the

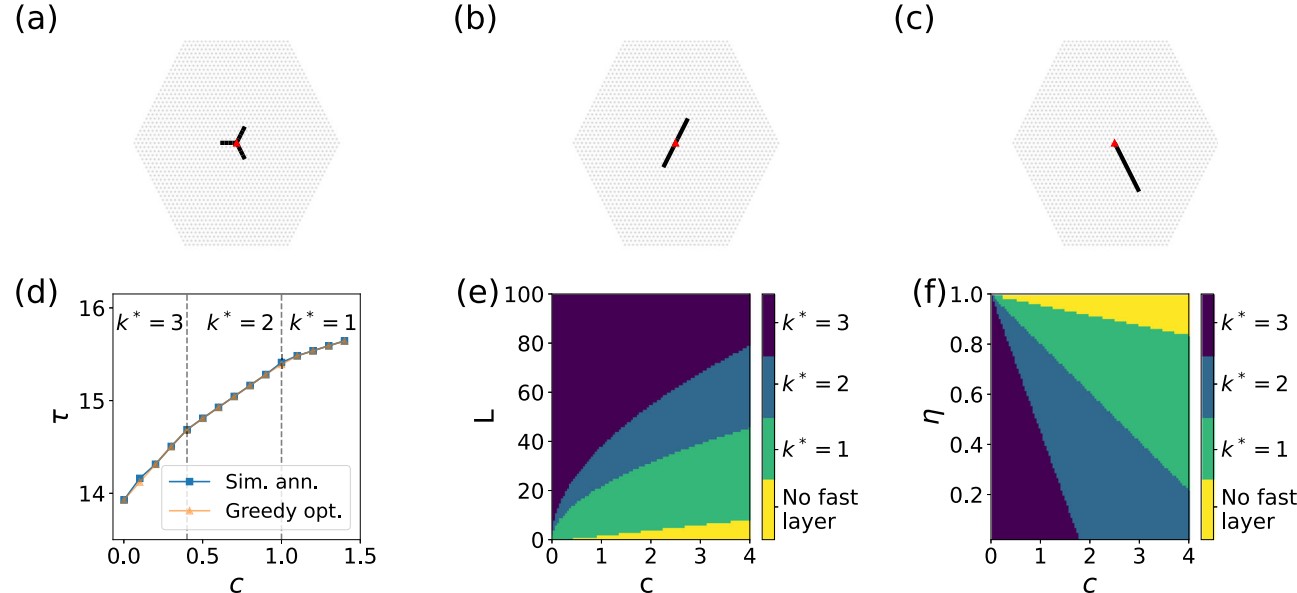

**Fig. 3 | Symmetry breaking in two-dimensional systems. a** Symmetric optimal configuration of the fast layer with $k^* = 3$ branches. Here $R = 25$ and $L = 12$. **b** Symmetric optimal configuration of the fast layer with $k^* = 2$ branches obtained for radius $R = 25$ and length of fast layer $L = 12$. **c** Asymmetric optimal configuration of the fast layer with $k^* = 1$ branch valid for $R = 25$ and $L = 12$. **d** Average time to the center, i.e., Eq. (1) associated with the optimal fast-layer configuration as a function of the switching cost $c$. Here, $R = 25, L = 12$ and the relative speed $\eta = 0.1$. We compare solutions obtained using greedy and simulated-annealing optimization. The vertical dashed lines correspond to the values of $c$ where we observe a change in the topology of the optimal fast layer. **e** Number of branches characterizing the topology of the optimal fast layer as a function of $L$ and $c$. Here, $R = 100$ and $\eta = 0.1$. **f** Same as in (**e**), but as a function of $\eta$ and $c$, with $L = 50$.

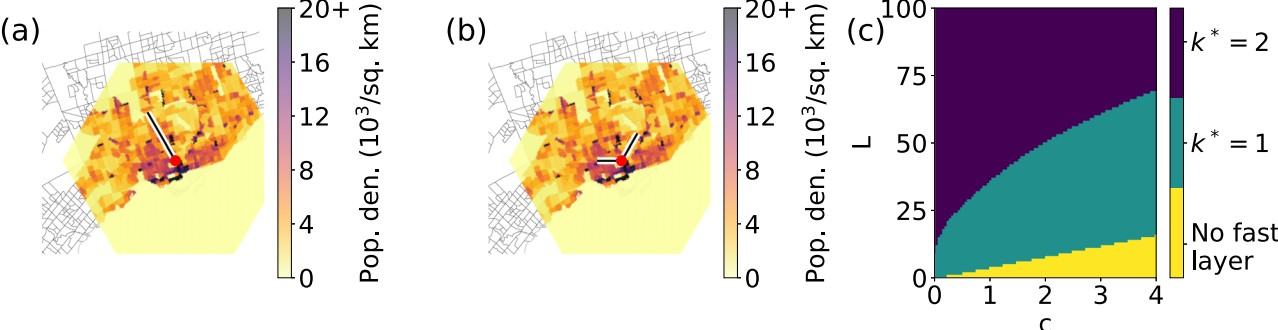

**Fig. 4 | Symmetry breaking in the transportation networks of real cities. a** We consider the city of Toronto. We construct the slow layer of the system using a triangular lattice radius $R = 100$; for the fast layer, we impose the length of the fast layer $L = 50$. The color map shows the population density associated with the lattice points; the gray lines represent census-tract boundaries. The red circle represents the center. We show the optimal configuration of the fast layer with $k^* = 1$ branch. (**b**) Same as in (**a**), but we show the solution with $k^* = 2$. **c** Phase diagram displaying the value of $k^*$ as a function of model parameters $L$ and $c$. Here the relative speed $\eta = 0.5$. The yellow region denotes $L \leq r_c$, where $r_c = 2c/(1 - \eta)$.

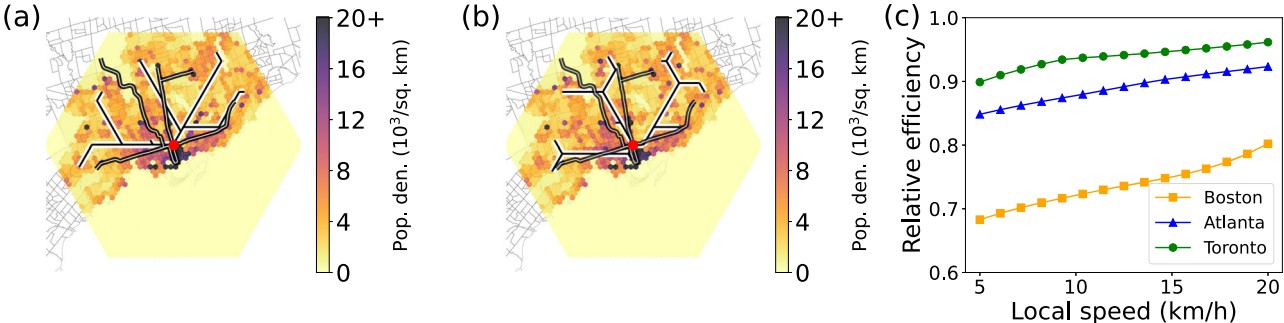

**Fig. 5 | Assessing the optimality of the transportation networks of real cities. a** We compare the subway network generated with our optimization framework (white curves) with the real subway system (black curves) in the city of Toronto. The optimized configuration is obtained by setting radius $R = 25$, relative speed $\eta = 0.5$, and switching cost $c = 1.25$ in the lattice model. These choices correspond to setting, in the physical system, the speed of the slow and fast layers respectively to $\bar{v}_s = 20$ km h$^{-1}$ and $\bar{v}_f = 40$ km h$^{-1}$, and the switching time between layers to 3 min. The latter setting means that 6 minutes are required to change mode of transportation, since the actual cost associated to the use of the fast layer is $2c$. **b** Same as in (**a**), but for $\bar{v}_s = 5$ km h$^{-1}$. We also set $c = 0.3125$ in the lattice model so that the switching time in the physical system still equals 3 mins. **c** The efficiency of the real subway systems relative to the optimal configurations as a function of the speed of the slow layer $\bar{v}_s$. Relative efficiency is given by the ratio between the values of the objective function estimated via Eq. (6) for the optimized and the real configurations of the fast layer. As we vary $\bar{v}_s$, we change also the value of the parameter $c$ in the multiplex model so that the switching time in the physical system is equal to 3 mins, see "Methods" for details.

radius of the city. Overlaying a city on top of the triangular lattice allows us to associate a weight $\tilde{p}_n$ to each node $n$ in the slow layer that reflects the real population density within the city. We use those weights in the objective function of Eq. (6), and then take advantage of the greedy algorithm to obtain approximate solutions to the optimization problem of Eq. (4). Similar to the previous sections, we obtain two classes of optimal fast-layer configurations for all the considered parameters. Results for the city of Toronto are displayed in Fig. 4, where we see that optimal configurations comprise $k^* = 1$ (Fig. 4a) or $k^* = 2$ (Fig. 4b) branches. Similar results are valid for Atlanta and Boston, where we observe optimal configurations with $k^* \leq 3$ branches (see Supplementary Information Figure 11). For $k^* > 1$, we note that the branches have no identical length; this is caused by the fact that the weight associated with the various nodes of the system is not constant. A typical phase diagram for Toronto is displayed in Fig. 4c, where we fix $\eta = 0.5$, but vary $L$ and $c$. The diagram is qualitatively similar to the one of Fig. 3e. For fixed $c$, $k^*$ increases as $L$ grows; however, for fixed $L$, $k^*$ decreases as $c$ grows. The values of the parameters where the transitions between the various phases emerge differ from those of Fig. 3e; this is due to the non-homogeneous density of the population used in the model of the city. Similar results for Atlanta and Boston can be found in the Supplementary Information Figure 11.

Next, we perform a direct comparison between the real subway lines and the optimal fast-layer configurations obtained using our computational framework. To this end, we calibrate the model's parameters $L$ and $R$ such that the number of subway stations in the real system is comparable with the one in the model. Results of this analysis are reported in Fig. 5 for Toronto and in the Supplementary Information Fig. 12 for Atlanta and Boston. We first note that the optimal fast-layer networks display additional ramifications. This is due to the fact that $L > R$ in this experimental setting. Second and more important, we note that there is an overall good overlap between the real subways and those obtained under the framework. This is true regardless of the specific choice of the model parameters (Fig. 5a, b). We quantify the efficiency of the real subway systems relative to the optimal configurations by measuring the ratio of the corresponding values of the objective function of Eq. (6), see Fig. 5c. Here, we keep the speed of the fast layer invariant as $\bar{v}_f = 40$ km h$^{-1}$, and vary the speed of the slow layer $\bar{v}_s$. This corresponds to effectively varying the value of the model parameter $\eta$. The real subway system appears less efficient than the optimized one in congested situations when $\bar{v}_s$ is small. However, it gets close to optimality as the speeds in the slow layer grows towards the value of the speed of the fast layer.

## Discussion

Location science and network design focused on practical aspects of network optimization. Even if operational research is successful in designing minimal cost solutions, the theoretical question of the optimal network topology is largely open. In addition, and as suspected in previous studies, we showed that these optimal networks experience a transition between different shapes when the total length or the switching cost increase: small variations of the cost can lead to strikingly dissimilar optimal structures. In particular, there is a transition characterized by a symmetry breaking leading to spatial inequalities. Such a phenomenon results from the interplay between switching cost and the absence of the network and shows that an optimal solution is not necessarily isotropic. Spatial inequality is known to happen in various economic instances[45]. It was also observed in the context of human mobility, e.g., car traffic[46–49], self-organized pedestrian movement[50,51], commuting patterns[52–54], and ride-sharing adoption[55,56]. It is, however, the first time that spatial inequality is exhibited in an optimal network context. Our results underscore the importance of considering switching costs and the cost associated with the slow layer (typically car traffic) when studying the optimal subway structures. A better theoretical understanding of these optimal shapes could certainly be helpful for practical applications and the identification of critical parameters. As optimal networks in transportation play a crucial role in ensuring efficient, safe, and sustainable mobility within urban (and regional) areas, we can expect our results to have some practical implications on various aspects of cities such as efficient mobility, cost-effectiveness, accessibility and connectivity, sustainability, resilience and adaptability, and economic development. In particular, optimal transportation networks support economic development by facilitating the efficient movement of goods and services, enhancing access to markets, and attracting investments and businesses to urban areas. Surprisingly enough, our findings show that efficient transportation networks do not cover a whole area, but rather focus on a smaller part of the area. This interplay between spatial coverage and access cost leads to this surprising result, especially in the light of general ideas about accessibility, well-connectedness that stimulate economic activity, create job opportunities, and improve the competitiveness of cities and regions. More generally, we can expect that for more complex objective functions – including resilience to urban growth for example – the optimal solution is not necessarily a spatially uniform one and we have to go beyond the paradigm of homogeneous access.

Further studies are however needed in order to explore in more depth these transitions. Also, we focused here on the monocentric case where we minimize the average distance to reach a central node. Large cities are however polycentric and the structure of flows is far more complex. Preliminary results suggest that here also there are transitions between different optimal shapes, but this point certainly deserves further studies. Finally, the more difficult problem of minimizing the average time needed to connect any pair of points is even more open. In this case, the optimal network can have loops and is computationally more demanding. Although we also expect transitions, our understanding of this case is at the beginning.

## Methods
### Multiplex transportation model
We consider a multiplex network composed of a slow layer and a fast layer (see Fig. 1). We denote with $\mathcal{G}$ the set of nodes in the slow layer, and with $\mathcal{S}$ the set of its edges; $\mathcal{H}$ and $\mathcal{F}$ are respectively the set of nodes and edges in the fast layer. Both layers contain $N$ nodes; each node $n$ in the slow layer has a one-to-one correspondence with a node $F(n)$ in the fast layer. Each edge $(F(n), F(m))$ in the fast layer has a replica edge $(n, m)$ in the slow layer, but the vice versa is not necessarily true. The transit time of each edge in the slow layer equals one, whereas time required to traverse edges of the fast layer is reduced by a factor

$0 \leq \eta \leq 1$. Replica nodes are connected to each other by edges whose transit time is $c \geq 0$. Please note that in this mathematical framework entities have no physical meaning, thus we can interchange the notions of the length of an edge with that of the time required to traverse it, and simply refer to them with the generic term cost.

When considered in isolation, the slow layer forms a single connected component, whereas the fast layer is not necessarily connected. The connectedness of the slow layer implies, however, that in the overall system, composed of the interconnected slow and fast layers, there exists at least a path connecting any pair of nodes. The cost of a path in the network is given by the sum of the costs of all edges that compose the path. The minimum-cost path between two nodes can either use edges in the slow layer only or take advantage of some of the edges in the fast layer (see Fig. 1). In particular, the path $n \to F(n) \to F(m) \to \ldots \to F(r) \to F(s) \to s$ composed of $\ell$ edges in the fast layer only is preferred to its replica path $n \to m \to \ldots \to r \to s$ whenever $\ell$ is larger than

$$r_c = \frac{2c}{1 - \eta} \, . \tag{3}$$

We identify a special node $o$ in the slow layer of the network, i.e., the center of the network. We then define the weighted average cost to the center as in Eq. (1). We stress that the objective function of Eq. (1) is computed over all nodes in the slow layer only, but eventual minimum-cost paths can take advantage of edges in the fast layer. Clearly, $\tau$ depends on the various parameters of the model. In Eq. (1), we explicit, on purpose, only the dependence of $\tau$ on the fast layer $\mathcal{F}$ as this is the primary object of our investigation. We consider in fact the optimization problem aimed at finding the best set of edges in the fast layer able to minimize the objective function of Eq. (1). The minimization is constrained by the number of edges $L$ that are in the fast layer, with $L$ still measured in the same units of costs as $\tau$ and $c$. Specifically, we aim at solving

$$\mathcal{F}^* = \arg\min_{|\mathcal{F}| = L} \tau(\mathcal{F}), \tag{4}$$

where we indicated with $|\mathcal{F}|$ the number of edges in $\mathcal{F}$.

Finding the exact solution to the optimization problem of Eq. (4) is computationally infeasible as it requires a brute-force search over all possible $\binom{|\mathcal{S}|}{L}$ configurations of the fast layer. In the SM, we prove, however, that: (i) the optimal configuration of the fast layer is a connected tree with at least one edge incident to $F(o)$, i.e., the replica node of the center; (ii) the objective function of Eq. (1) is a decreasing and submodular function. The relevance of property (i) is two-fold: first, it allows us to dramatically reduce the number of suitable solutions for the optimization problem of Eq. (4); second, it permits us to meaningfully describe the geometry of the optimal fast layer in terms of the number of branches departing from the replica node of the center, i.e.,

$$k^* = \sum_{(F(n), F(m)) \in \mathcal{F}^*} \left[ \delta_{F(o), F(m)} + \delta_{F(n), F(o)} \right], \tag{5}$$

where $\delta_{x,y} = 1$ if $x = y$ and $\delta_{x,y} = 0$ otherwise. Property (ii) allows us to leverage a greedy optimization scheme to generate approximate solutions to the optimization problem of Eq. (4) that are at most a fraction $(1 - 1/e)$ above the ground-truth minimum[44]. In the construction of greedy solutions, we start from an empty set of edges in the fast layer, and we add one edge at a time. The edge that is added is the one corresponding to the best choice that can be made given the current set of edges in the fast layer. In the SM, we further describe how solutions obtained via greedy optimization can be further refined to get better approximations for the optimization problem of Eq. (4); the quasi-optimality of our greedy solutions is validated by comparing

them to those obtained via simulated-annealing optimization (see Fig. 3 and SM for details).

## Two-dimensional triangular lattices

The slow layer used in the definition of the multiplex transportation model can be represented by any connected network. In the SM for example, we report on results obtained for a slow layer given by an instance of an Erdős-Rényi model.

The vast majority of the results reported in this paper are obtained for slow layers derived from triangular lattices. The coordinates of all sites of the lattice are in the form $(a + \frac{b}{2}, \frac{\sqrt{3}b}{2})$ for integer values of $a$ and $b$ such that $|a| + |b| \leq R$, where $R$ is the radius of the triangular lattice. The boundary conditions give the system a hexagonal shape. The sites of the lattice are the nodes of the slow layer; each pair of nodes in the slow layer is connected if the corresponding sites are at distance one in the triangular lattice; we identify the center of the network as the site with coordinates $(0, 0)$, i.e., $a = b = 0$.

## Real-world cities

In this section, we describe how we model the transportation system of a real city. Our framework relies on the use of a multiplex network formed of two discrete triangular lattices, one used to describe slow transportation (e.g., cars) and the other used to model fast transportation (e.g., subways). As we are referring to a real physical system, all quantities that describe properties of the multiplex transportation model have an associated physical dimension. For simplicity, we still rely on the same notation as in the previous sections, however, we add a tilde on the top of the symbols to make clear that the notation is used to indicate physical quantities. For example, we use $\tilde{R}$ to denote the city radius measured in units of length and distinguish it from $R$ which serves to indicate the radius of the triangular lattice measured in dimensionless lattice units.

We obtain the population density at the census-tract level for Boston and Atlanta from the 2021 American Census Survey, and for Toronto from the 2021 Canadian Census of Population. The census data contains the number of individuals residing in relatively small geographic regions, i.e., census tracts, used for statistical purposes by national statistical agencies. Census tracts typically consist of 2500 to 8000 individuals.

The data on the four metro lines and their stations in Boston is obtained from the Massachusetts Bureau of Geographic Information (MassGIS) website. Similar data for the four metro lines in Atlanta is obtained from the Atlanta Regional Commission: Open Data website. Finally, the data for the three subway lines and their stations in Toronto is made available by the Toronto Transit Commission at the City of Toronto Open Data website. The data on the subway lines is available as shapefiles. The arcs for the rail lines are given by sets of points in the coordinate reference systems (CRS) applicable to the geographic location. The CRSs used for Atlanta, Boston, and Toronto are EPSG:2239, EPSG:26986, and, EPSG:2952, respectively. Similarly, the location of the stations is given by points in the appropriate CRS. The Atlanta, Boston, and Toronto subway systems total $\tilde{L} = 77$ km, $\tilde{L} = 109.6$ km, and $\tilde{L} = 69.6$ km of rail lines and $\tilde{n}_s = 38, \tilde{n}_s = 125$, and $\tilde{n}_s = 75$ stations, respectively. The average distance between the stations is 2.07 km, 0.88 km, and 0.94 km for Atlanta, Boston, and Toronto, respectively.

We choose a city radius $\tilde{R}$ such that all stations are contained in the circle of radius $\tilde{R}$ around the center. We find the appropriate choice for this radius to be $\tilde{R} = 25$ km in Atlanta and $\tilde{R} = 20$ km in Boston and Toronto. We assume that everything that lies inside this circular area constitutes the city.

We create a lattice model of the transportation system in a city by overlaying a triangular lattice of radius $R$ on top of the circle of radius $\tilde{R}$. Please note that we use $R = 100$ in all figures except for Fig. 5 where we use $R = 25$. The operation requires matching locations of the city

that are given by points in continuous space to lattice sites. To this end, we fix the location of the city center as the one of the center $o$ of the triangular lattice. Since the lattice covers the entire circular area of the city, the physical distance between neighboring sites on the lattice is $\tilde{w}_{n,m} = \tilde{R}/R$. The choice of $R$, for given $\tilde{R}$, determines the granularity of the lattice mesh overlaid on the city landscape. For instance, $R = 100$ and $\tilde{R} = 25$ km give us neighboring lattice sites $n$ and $m$ at distance $\tilde{w}_{n,m} = 0.25$ km, whereas $R = 25$ and $\tilde{R} = 25$ km give $\tilde{w}_{n,m} = 1$ km. Note that the physical distance between nodes $n$ and $m$ in the two layers is the same as the physical distance between their replica nodes $F(n)$ and $F(m)$ in the fast layer, i.e., for all $(F(n), F(m)) \in \mathcal{F}$ $\tilde{w}_{F(n),F(m)} = \tilde{w}_{n,m}$. The distance between the replica nodes $n$ and $F(n)$ is a parameter of the model $\tilde{w}_{n,F(n)} = \tilde{\ell}$.

The weight $\tilde{p}_n$ associated with node $n$ in the slow layer is given by the population density of the associated census tract containing the site. Note that the size of census tracts varies significantly as the population density varies. Consequently, depending on the choice of $R$ and $\tilde{R}$, we may have no lattice sites in very small census tracts. We find that this issue can be resolved by choosing $R = 100$ for the values of $\tilde{R}$ indicated above.

We assume that the travel speed $\tilde{v}_s$ on the slow layer is between $5 \text{ km h}^{-1}$ and $20 \text{ km h}^{-1}$, and the speed on the fast layer is $\tilde{v}_f = 40 \text{ km h}^{-1}$. These are both realistic (ranges of) values for the travel speeds of cars and subways, respectively. Clearly, we have $\eta = \tilde{v}_s/\tilde{v}_f$. We note that the time required to traverse the edge $(n, m) \in \mathcal{S}$ is $\tilde{t}_{n,m} = \tilde{w}_{n,m}/\tilde{v}_s$, whereas the time required to traverse the edge $(F(n), F(m)) \in \mathcal{F}$ is $\tilde{t}_{F(n),F(m)} = \eta \tilde{t}_{n,m}$. For example, if $R = 25, \tilde{R} = 25$ km, $\tilde{v}_s = 20 \text{ km h}^{-1}$, and $\tilde{v}_f = 40 \text{ km h}^{-1}$, we have $\tilde{t}_{n,m} = \frac{1}{20}$ hours or 3 min, and $\tilde{w}_{F(n),F(m)} = \frac{1}{40}$ h or 1.5 min. Finally, we assume that a change of layers occurs also at speed $\tilde{v}_s$, meaning that the time required to switch layers is $\tilde{t} = \tilde{\ell}/\tilde{v}_s$. In our simulations, we impose $\tilde{t} = 3$ mins. This choice means that 6 mins are required to change mode of transportation, since the actual cost associated to the use of the fast layer is $2\tilde{t}$.

We denote with $\tilde{t}_n$ the time required to reach the center $o$ from node $n$. This is given by the time corresponding to the fastest path connecting the two nodes. We finally rewrite Eq. (1) as

$$\tilde{\tau}(\mathcal{F}) = \frac{\sum_{n \in \mathcal{G}} \tilde{t}_n \tilde{p}_n}{\sum_{n \in \mathcal{G}} \tilde{p}_n}, \tag{6}$$

and use this expression while solving the optimization problem of Eq. (4).

Next, we explain the modeling framework used to obtain the results for the real subway systems in Fig. 5. The slow layer is modeled identically as described above. We use $\tilde{n}_s$ stations as the nodes of the fast layer. Connections between stations are given by actual railways, with length learned directly from the data. For each station, we identify its replica in the slow layer as the node corresponding to the closest (in terms of geographical distance) site of the triangular lattice.

To properly compare the objective function of Eq. (6) for the real city against the one obtained after greedy optimization, we set $R = 25$ when constructing the triangular lattice. This allows us to obtain comparable numbers of subway stations between the real cities and the synthetic ones. We respectively have $L = 77, 132$, and 87 for the synthetic versions of Atlanta, Boston, and Toronto.

## Continuous-space approximation for one-dimensional lattices

To ease analytical calculations, we adopt a continuous-space approximation for a multiplex transportation model where the slow layer is given by a one-dimensional lattice. Here for simplicity, we assume that the weight associated to each node $n$ in the slow layer is $p_n = \text{const}$. In the SM, we prove that the symmetry breaking occurs for arbitrary nonnegative functions $p_n$ that are symmetric with respect to the center of the lattice. Under the continuous-space approximation, the slow layer has the center located in the origin, and is formed of two segments of

length $R$ extending symmetrically to the left and the right of the center (see Fig. 2a). The fast layer extends to the right of the origin with a segment of length $\alpha L$ and to the left with a segment of length $(1-\alpha)L$, where $0 \le \alpha \le 1/2$ is a tunable parameter and $L \le R$ is the total length of the fast layer. The goal of our calculation is to find the value $\alpha^*$ corresponding to the optimal configuration of the fast layer, i.e., the solution of the continuous-space approximation of the optimization problem of Eq. (4).

Consider first the right side of the fast layer which is serving the right side of the slow layer. The objective function relative to the right side of the system is

$$\tau_{\text{right}}(\alpha) = \begin{cases} R^2/2 & \text{if } 0 \le \alpha \le r_c/L, \\ C + (1-\eta)\,L\left[\alpha^2 L/2 - \alpha R\right] & \text{oth.} \end{cases} \tag{7}$$

where

$$C = \frac{1}{2}(1-\eta)r_c^2 + 2c(R - r_c) + \frac{1}{2}R^2. \tag{8}$$

If $\alpha L < r_c$, the fast layer does not serve any portion of the slow layer, thus $\tau_{\text{right}}(\alpha) = \int_0^R dx\, x = R^2/2$. If $\alpha L \ge r_c$, we need to solve the integral $\tau_{\text{right}}(\alpha) = \int_0^{r_c} dx\, x + \int_{r_c}^{R} dx\,(2c + \eta x) + \int_{\alpha L}^{R} dx\,(2c - (1-\eta)\alpha L + x)$, leading to the second case of Eq. (7). We note that the term $C$ appearing in Eq.(8) does not depend on $\alpha$, but only on $r_c$ and $R$.

For the left portion of the fast layer, we simply have $\tau_{\text{left}}(\alpha) = \tau_{\text{right}}(1-\alpha)$. For the entire system, the objective function reads $\tau(\alpha) = \tau_{\text{right}}(\alpha) + \tau_{\text{left}}(\alpha)$.

We now distinguish two cases: (i) $r_c/L \le 1/2$ and (ii) $r_c/L \ge 1/2$. In case (i), we can write:

$$\tau(\alpha) = \begin{cases} R^2/2 + C + (1-\eta)\,L\left[(1-\alpha)^2 L/2 - (1-\alpha)R\right] & \text{if } 0 \le \alpha \le r_c/L \\ 2C + (1-\eta)\,L\left[((1-\alpha)^2 + \alpha^2)L/2 - R\right] & \text{if } r_c/L \le \alpha \le 1/2. \end{cases} \tag{9}$$

thus,

$$\frac{d\tau(\alpha)}{d\alpha} = \begin{cases} (1-\eta)\,L(R - (1-\alpha)L) \ge 0 & \text{if } 0 \le \alpha \le r_c/L \\ (1-\eta)\,L[-L - R] \le 0 & \text{if } r_c/L \le \alpha \le 1/2. \end{cases}$$

We see therefore that the function reaches its maximum at $\alpha = r_c/L$, and displays its minimum value either in $\alpha = 0$ or $\alpha = 1/2$. To determine where the minimum of the objective function of Eq. (9) is obtained, we need to solve the equation $\tau(\alpha = 0) = \tau(\alpha = 1/2)$. After some simple calculations, we arrive to

$$r^\dagger = R\left[1 - \sqrt{1 - \frac{1}{2}\left(\frac{L}{R}\right)^2}\right]. \tag{10}$$

For $r_c \ge r^\dagger$ the optimal configuration is the one obtained for $\alpha^* = 1/2$, whereas for $r_c \le r^\dagger$ the optimal configuration is the one corresponding to $\alpha^* = 0$.

In case (ii), we can repeat a similar derivation. We find that the maximum of the objective function is reached in $\alpha = 1 - r_c/L$, and the function displays its minimum value either in $\alpha = 0$ or $\alpha = 1/2$. Also, here we find the critical value of Eq. (10) where the optimal configuration of the fast layer changes from being perfectly symmetric to being asymmetric. Alternatively, we can determine the critical length $L^\dagger$ of the fast layer as shown in Eq. (2).

## Reporting summary

Further information on research design is available in the Nature Portfolio Reporting Summary linked to this article.

## Data availability

The datasets analyzed during the current study are available at: Census Reporter, https://censusreporter.org/, Toronto Transit Commission, https://open.toronto.ca/dataset/ttc-subway-shapefiles/, Atlanta Regional Commission, https://opendata.atlantaregional.com/datasets/09f07db2c25c41db945369f050a87bf5_0/explore, and MassGIS, Boston rail data, https://www.mass.gov/info-details/massgis-data-trains.

## Code availability

The code developed for this paper is available at https://zenodo.org/records/10659254.

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

## Acknowledgements

This project was partially supported by the Army Research Office under contract number W911NF-21-1-0194, by the Air Force Office of Scientific Research under award numbers FA9550-19-1-0391 and FA9550-21-1-0446, and by the National Science Foundation under award number 1927418. The funders had no role in study design, data collection, and analysis, the decision to publish, or any opinions, findings, conclusions, or recommendations expressed in the manuscript.

## Author contributions

S.P. and F.R. performed the experiments. S.P., M.B., S.E., S.F., and F.R. conceived and designed the experiments and wrote the paper.

## Competing interests
The authors declare no competing interests.
