## [Peer Review File · Nature Communications]

REVIEWER COMMENTS

Reviewer #1 (Remarks to the Author):

The manuscript "Symmetry breaking in optimal transport networks" studies how additional, faster layers of transport may be added to an existing slow-mode layer such as to optimize the total perceived time to reach a predefined central node.

noteworthy results:

 The authors uncover a symmetry breaking mechanism that they fully explain for one-dimensional systems. They also explore two-dimensional as well as in an infinite-dimensional settings.

significance to the field and related fields/ How does it compare to the established literature?

 To the best of my knowledge, the work's main setting and results are novel. They offer a new way of considering the design of optimal transport networks in principle. As stated, the claim that symmetry breaking occurs in optimal network context overall seems an overstatement, see, .e.g. Storch et al, Nature Comm. 2021, an explicit work on symmetry breaking by Wolf et al., <https://arxiv.org/abs/2103.16891>, both on ride sharing, as well as references therein.

I think it makes sense the authors of the current manuscript sharpen their statement on what is truly novel.

Does the work support the conclusions and claims, or is additional evidence needed? Are there any flaws in the data analysis, interpretation and conclusions? Do these prohibit publication or require revision? Is the methodology sound? Does the work meet the expected standards in your field? Is there enough detail provided in the methods for the work to be reproduced?

The work seems sound and the conclusions justified. I did not fully understand the meaning of the p_n in eqn (3) which weigh the individual costs.

Moreover, while the authors have applied their approach to data from real cities, it is not clear to which extent the results may help the design of real transport networks beyond providing the information that symmetry breaking may exist.

I believe the work is publishable with minor revisions but I am not sure given the current manuscript how significant the results are for actual applications. Specifically, cities are not optimized for reaching one central point but rather (ideally) for many people reaching their diverse destinations from the diverse origins of travel. To me it seems an open question whether the insights provided here would generalize to such settings, even in a theoretical model.

Reviewer #2 (Remarks to the Author):

The topic is very important, one result from Phys Rev Lett that inspired real world applications, is this one:

Transport on coupled spatial networks

RG Morris, M Barthelemy

Physical review letters 109 (12), 128703

Why that work is not cited here?

I value the theoretical approach but the authors must do the effort to connect their world to real world data.

Reviewer #3 (Remarks to the Author):

Report on "Symmetry Breaking in Optimal Transport Networks"

The manuscript presents an interesting study on the optimal shape of multiplex networks for efficiently connecting a set of points, a question of significant theoretical and practical relevance. The authors propose a model based on a multiplex network framework, where a fast layer is embedded within a slower one, allowing for the exploration of different strategies to minimize the average time to reach a

central node across various dimensions (d). The consideration of a switching cost between layers introduces a distinctive layer to the analysis, reflecting real-world constraints in network design.

By analyzing the case of $d = 1$ analytically and cases of $d > 1$ numerically, the authors reveal transitions in the optimal network structure influenced by the network size, the switching cost, and the relative speeds of the two layers. The manuscript highlights an interesting counterintuitive finding that sometimes it is beneficial to exclude certain areas from the fast layer (which they call breaking of symmetry) to save on switching costs, even if it means relying more on the slower layer. This result emphasizes the trade-offs in network design, particularly the role of switching costs in determining the optimal network configuration. Applying theoretical findings to real-world subway systems in cities like Atlanta, Boston, and Toronto, the manuscript promotes a plausible connection between abstract network models and tangible urban transport systems. This comparison illustrates how real-world systems diverge from theoretically optimal networks, particularly as traffic congestion intensifies.

The manuscript introduces the perspective on network optimization to focus on the role of switching costs and the phenomenon of symmetry breaking. However, it would greatly benefit from a more extensive comparison with existing models in network theory and optimization. Understanding how these new findings align with or diverge from established theories should be discussed since it would provide a deeper insight into the significance and applicability of the research. In this context, I suggest that the authors will add to their discussion two previous relevant works on navigation in transport networks, namely, "Navigation in a small world", by J. M. Kleinberg, *Nature (London)* 406, 845 (2000) and "Towards Design Principles for Optimal Transport Networks", by G. Li et al., *Phys. Rev. Lett.* 104, 018701 (2010). In particular, the PRL paper deals with a similar (but not the same) optimization for multiplexes in $2d$.

Finally, overall the idea presented is interesting but the manuscript suffers from issues of clarity in its methodological description—even Fig. 2a is not easy to understand. The presentation of the model and its underlying equations is somewhat disorganized, with key equations and model characteristics introduced only after being referred in the text. This approach significantly hinders the reader's ability to comprehend the methodology and follow the logical progression of the analysis. A more structured and clear presentation, with a logical flow from the introduction of the model to the presentation of findings, would greatly enhance readability and comprehension. I also suggest the authors to justify the cost of transitions since I think it might be more realistic to in the fast speed according to the the use --e.g., number of kilometers.

I believe that after the above suggested appropriate revisions made by the authors, the manuscript would fit to be published in *Nature Communications*.

Referee 1

The manuscript "Symmetry breaking in optimal transport networks" studies how additional, faster layers of transport may be added to an existing slow-mode layer such as to optimize the total perceived time to reach a predefined central node.

Noteworthy results: The authors uncover a symmetry breaking mechanism that they fully explain for one-dimensional systems. They also explore two-dimensional as well as in an infinite-dimensional settings.

We thank the reviewer for the time dedicated to our manuscript and for the very positive assessment of our contribution.

Significance to the field and related fields/ How does it compare to the established literature?

To the best of my knowledge, the work's main setting and results are novel. They offer a new way of considering the design of optimal transport networks in principle. As stated, the claim that symmetry breaking occurs in optimal network context overall seems an overstatement, see, .e.g. Storch et al, Nature Comm. 2021, an explicit work on symmetry breaking by Wolf et al., both on ride sharing, as well as references therein. I think it makes sense the authors of the current manuscript sharpen their statement on what is truly novel.

We thank the reviewer for this comment. We included the above papers, and several of the references therein, to the list of references of the revised version of the paper. In particular, we modified the discussion of the revised version of the paper to give proper credit to the existing literature on symmetry-breaking phenomena in human mobility.

Does the work support the conclusions and claims, or is additional evidence needed? Are there any flaws in the data analysis, interpretation and conclusions? Do these prohibit publication or require revision? Is the methodology sound? Does the work meet the expected standards in your field? Is there enough detail provided in the methods for the work to be reproduced?

The work seems sound and the conclusions justified. I did not fully understand the meaning of the p_n in eqn (3) which weigh the individual costs.

Moreover, while the authors have applied their approach to data from real cities, it is not clear to which extent the results may help the design of real transport networks beyond providing the information that symmetry breaking may exist.

I believe the work is publishable with minor revisions but I am not sure given the current manuscript how significant the results are for actual applications. Specifically, cities are not optimized for reaching one central point but rather (ideally) for many people reaching their diverse destinations from the diverse origins of travel. To me it seems an open question whether the insights provided here would generalize to such settings, even in a theoretical model.

p_n indicates the weight of node n in the calculation of the objective function of Eq.(3). This is interpreted as the number of people that want to go from node n to the center of the network. For instance, in the analysis of real cities, p_n is set equal to the empirical population density of the corresponding census tract. All main theoretical calculations of the original version of the paper assumed $p_n = \text{const}$. However, stimulated by the comment by the reviewer, we expanded our theoretical analysis for the one-dimensional lattice. We could prove that all findings hold for arbitrary non-negative functions p_n that are symmetric with respect to the center of the lattice. We included the new calculations in the revised version of the SM and mentioned these new results in the revised version of the main paper.

We agree with the reviewer that assuming that there is only one center in a city is a strong assumption valid (to some extent) only for some real cities. The cities that were considered in our empirical analysis were indeed selected for their monocentric structure. These limitations are clearly stated in the final part of the discussion section, i.e., “Also, we focused here on the monocentric case where ... expect transitions, our understanding of this case is at the beginning.”

Once more, we thank the reviewer for the time dedicated to our work and for the endorsement of its publication in the journal.

Referee 2

The topic is very important, one result from Phys Rev Lett that inspired real world applications, is this one:
Transport on coupled spatial networks RG Morris, M Barthelemy Physical review letters 109 (12), 128703
Why that work is not cited here?
I value the theoretical approach but the authors must do the effort to connect their world to real world data.

We thank the reviewer for the time dedicated to our manuscript and for the positive response.

We included a citation to the above reference in the revised version of the paper.

Concerning the main criticism by the reviewer, we recognize that our contribution is primarily theoretical and computational. However, we believe that some of the methods introduced in the paper are already sufficiently applicable to real-world data. For example, the greedy algorithm proposed in the paper allows us to generate fast-layer networks that have a guaranteed bound on their optimality due to the submodularity property of the objective function that we prove in the SM. We applied the greedy algorithm to three real cities, relying on publicly available data about their population density. The results show a qualitatively good agreement between our optimal solutions and the real structure of the subway lines of these cities (see Fig. 5 of the main paper and Fig. S12 in the SM). The similarity between the computational model and the real network is obtained in spite of the fact that our definition of optimality relies on a monocentric assumption that is not necessarily realistic.

Referee 3

The manuscript presents an interesting study on the optimal shape of multiplex networks for efficiently connecting a set of points, a question of significant theoretical and practical relevance. The authors propose a model based on a multiplex network framework, where a fast layer is embedded within a slower one, allowing for the exploration of different strategies to minimize the average time to reach a central node across various dimensions (d). The consideration of a switching cost between layers introduces a distinctive layer to the analysis, reflecting real-world constraints in network design.

By analyzing the case of $d = 1$ analytically and cases of $d > 1$ numerically, the authors reveal transitions in the optimal network structure influenced by the network size, the switching cost, and the relative speeds of the two layers. The manuscript highlights an interesting counterintuitive finding that sometimes it is beneficial to exclude certain areas from the fast layer (which they call breaking of symmetry) to save on switching costs, even if it means relying more on the slower layer. This result emphasizes the trade-offs in network design, particularly the role of switching costs in determining the optimal network configuration. Applying theoretical findings to real-world subway systems in cities like Atlanta, Boston, and Toronto, the manuscript promotes a plausible connection between abstract network models and tangible urban transport systems. This comparison illustrates how real-world systems diverge from theoretically optimal networks, particularly as traffic congestion intensifies.

We thank the reviewer for the time dedicated to our manuscript and for the very positive report.

The manuscript introduces the perspective on network optimization to focus on the role of switching costs and the phenomenon of symmetry breaking. However, it would greatly benefit from a more extensive comparison with existing models in network theory and optimization. Understanding how these new findings align with or diverge from established theories should be discussed since it would provide a deeper insight into the significance and applicability of the research. In this context, I suggest that the authors will add to their discussion two previous relevant works on navigation in transport networks, namely, "Navigation in a small world", by J. M. Kleinberg, *Nature* (London) 406, 845 (2000) and "Towards Design Principles for Optimal Transport Networks", by G. Li et al., *Phys. Rev. Lett.* 104, 018701 (2010). In particular, the PRL paper deals with a similar (but not the same) optimization for multiplexes in 2d.

We added a new sentence in the introduction of the revised version of the manuscript to address the comment by the referee. We also included citations to the above papers.

Finally, overall the idea presented is interesting but the manuscript suffers from issues of clarity in its methodological description—even Fig. 2a is not easy to understand. The presentation of the model and its underlying equations is somewhat disorganized, with key equations and model characteristics introduced only after being referred in the text. This approach significantly hinders the reader’s ability to comprehend the methodology and follow the logical progression of the analysis. A more structured and clear presentation, with a logical flow from the introduction of the model to the presentation of findings, would greatly enhance readability and comprehension. I also suggest the authors to justify the cost of transitions since I think it might be more realistic to in the fast speed according to the use –e.g., number of kilometers. I believe that after the above suggested appropriate revisions made by the authors, the manuscript would fit to be published in Nature Communications.

Thanks for making these comments. They helped us to greatly improve the readability of the paper.

As noted by the reviewer, the old version of Fig. 2a was unclear. We regenerated the figure in a format consistent with that of Fig. 1. The caption of the figure has been updated accordingly.

Regarding the organization of the paper, we followed the typical structure of papers appearing in this journal, where technical details are generally reported in the Methods section appearing at the end of the paper, see <https://www.nature.com/ncomms/submit/article>. We agree with the reviewer that this type of organization of the material may not be the best for a reader expert in the subject of the paper. On the other hand, it allows grasping the main messages of the paper without the need of first reading long technical sections. The latter scenario could be preferred by a reader who is not a specialist in the topic.

Finally, the switching cost on empirical networks is reported in units of time. This is done for consistency as this is the same physical dimension associated with the objective function. Please note that, however, we tune the switching cost by effectively changing the length of the edges between nodes in the slow layer and their replicas in the fast layer. As explained in the Methods, several physical quantities are fixed based either on constraints or realistic modeling assumptions (e.g., size of the city, typical speeds of cars and subways). Specifically, “we assume that a change of layers occurs also at speed \tilde{v}_s , meaning that the time required to switch layers is $\tilde{\tau} = \tilde{\ell}/\tilde{v}_s$.” Here, \tilde{v}_s is the speed in the slow layer, and $\tilde{\ell}$ the length of inter-layer edges. We added new sentences in the caption of Figure 5 and in the Methods section of the revised version of the paper to better clarify this modeling aspect.

List of Changes

To ease the re-review of the manuscript, we highlighted major changes with red fonts. We performed the following changes to the manuscript:

1. We updated Fig.2a to address a comment by reviewer 3.
2. We added a new set of calculations to the SM. Those calculations are also commented in the Methods section of the main paper. The calculations were prompted by a comment of reviewer 1.
3. We included in the introduction and in the discussion a few additional paragraphs aimed at addressing comments about missing references by reviewers 1, 2 and 3.
4. We added the sections “Data availability,” “Code availability,” and “Author contributions” to the revised version of the paper.

REVIEWER COMMENTS

Reviewer #1 (Remarks to the Author):

The authors have revised their manuscript in response to the comments by three reviewers.

Many of the issues raised in the reviews have been addressed by reworking parts of the manuscript, and overcome or circumvented.

I still believe that the current manuscript can be further improved regarding its presentation to improve the accessibility of the work and to emphasize the work's main message.

Generally, I follow the view and suggestion of reviewer 3 that the logical flow can be improved to help the readers walk through the manuscript's main statements and steps. I believe the meaning of the weights p_n should be introduced on first appearance and that the measure of efficiency defined in eqn. (3) should be defined earlier in the ms., I think on page 2. That may in particular help to improve readability and avoid asking the reader to refer to material later in the manuscript to understand the logical flow of the work. Finally, I believe it might be helpful if the authors more specifically discuss the potential implications of their findings for real cities. Given that they obtained some generally valid findings (e.g., the counterintuitive result that excluding certain areas from the fast layer may be beneficial as this might save switching costs) and do apply their approach on real city data, discussing general implications along these lines may help improving the accessibility and reach of the current work.

Reviewer #2 (Remarks to the Author):

I think the authors successfully addressed the comments by me and the other two referees. I think the work is suitable for publication.

Reviewer #3 (Remarks to the Author):

The authors considered well my comments in the earlier version and the current version of the revised manuscript is suitable for publication in Nature Communications

Referee 1

The authors have revised their manuscript in response to the comments by three reviewers.

Many of the issues raised in the reviews have been addressed by reworking parts of the manuscript, and overcome or circumvented.

We thank the reviewer for the time dedicated to our manuscript and for the positive assessment of our contribution.

I still believe that the current manuscript can be further improved regarding its presentation to improve the accessibility of the work and to emphasize the work's main message.

We address the final comments of the reviewer in revised version of the paper, see below for details.

Generally, I follow the view and suggestion of reviewer 3 that the logical flow can be improved to help the readers walk through the manuscript's main statements and steps. I believe the meaning of the weights p_n should be introduced on first appearance and that the measure of efficiency defined in eqn. (3) should be defined earlier in the ms., I think on page 2. That may in particular help to improve readability and avoid asking the reader to refer to material later in the manuscript to understand the logical flow of the work.

We followed the suggestion by the reviewer and placed the definition of the objective function in the results section of the main manuscript (i.e., page 2). We also provided an explanation of the weights p_n as well as explicit their meaning in the context of real cities.

Finally, I believe it might be helpful if the authors more specifically discuss the potential implications of their findings for real cities. Given that they obtained some generally valid findings (e.g., the counterintuitive result that excluding certain areas from the fast layer may be beneficial as this might save switching costs) and do apply their approach on real city data, discussing general implications along these lines may help improving the accessibility and reach of the current work.

We thank the referee for this suggestion. We enriched the discussion about the general and practical implications of our findings.

Referee 2

I think the authors successfully addressed the comments by me and the other two referees. I think the work is suitable for publication.

We thank the referee for the time spent on our manuscript and for the positive report.

Referee 3

The authors considered well my comments in the earlier version and the current version of the revised manuscript is suitable for publication in Nature Communications

We thank the referee for the time spent on our manuscript and for the positive report.

List of Changes

To ease the re-review of the manuscript, we highlighted major changes with red fonts. We performed the following changes to the manuscript:

1. We addressed the first comment by referee 1 and included the definition of the objective function at the beginning of the section "Results."
2. We addressed the second comment by referee 1 and expanded the discussion part of the paper to highlight implications of our findings for real transportation systems.